# Hydrothermal Synthesis of Functionalized Carbon Nanodots and Their Clusters as Ionic Probe for High Sensitivity and Selectivity for Sulfate Anions with Excellent Detection Level

**DOI:** 10.3390/polym15122655

**Published:** 2023-06-12

**Authors:** Po-Chih Yang, Pradeep Kumar Panda, Cheng-Han Li, Yu-Xuan Ting, Yasser Ashraf Gandomi, Chien-Te Hsieh

**Affiliations:** 1Department of Chemical Engineering and Materials Science, Yuan Ze University, Taoyuan 32003, Taiwan; rkpanda277@gmail.com (P.K.P.); s1071127@mail.yzu.edu.tw (C.-H.L.); danny0935331003@gmail.com (Y.-X.T.); 2Department of Chemical Engineering, Massachusetts Institute of Technology, Cambridge, MA 02142, USA; ygandomi@mit.edu

**Keywords:** carbon nanodots, nitrogen functionalization, hydrothermal synthesis photoluminescence quenching, ionic recognition

## Abstract

Nitrogen-doped carbon nanodots (CNDs) were synthesized and utilized as sensing probes to detect different anions and metallic ions within aqueous solutions. The pristine CNDs were developed through a one-pot hydrothermal synthesis. *o*-Phenylenediamine was used as the precursor. A similar hydrothermal synthesis technique in the presence of polyethylene glycol (PEG) was adopted to form the PEG-coated CND clusters (CND-100k). Through photoluminescence (PL) quenching, both CND and PEG-coated CND suspensions display ultra-high sensitivity and selectivity towards HSO_4_^−^ anions (Stern–Volmer quenching constant (*K*_SV_) value: 0.021 ppm^−1^ for CND and 0.062 ppm^−1^ for CND-100k) with an ultra-low detection limit (LOD value: 0.57 ppm for the CND and 0.19 ppm for CND-100k) in the liquid phase. The quenching mechanism of N-doped CNDs towards HSO_4_^−^ ions involves forming the bidentate as well as the monodentate hydrogen bonding with the sulfate anionic moieties. The detection mechanism of metallic ions analyzed through the Stern–Volmer formulation reveals that the CND suspension is well suited for the detection of Fe^3+^ (*K*_SV_ value: 0.043 ppm^−1^) and Fe^2+^ (*K*_SV_ value: 0.0191 ppm^−1^) ions, whereas Hg^2+^ (*K*_SV_ value: 0.078 ppm^−1^) sensing can be precisely performed by the PEG-coated CND clusters. Accordingly, the CND suspensions developed in this work can be employed as high-performance PL probes for detecting various anions and metallic ions in the liquid phase.

## 1. Introduction

Carbon nanodots (CNDs) and graphene quantum dots (GQDs) are promising nanomaterials that are commonly synthesized through the carbonization of hydrocarbon or graphene-based precursors [1,2,3,4]. Typically, the average particle size of CNDs and GQDs is less than 10 nm. Considering the quantum size confinement effect, their photoluminescence (PL) properties can be finely adjusted for variable wavelength and photobleaching impedance [5]. Compared to traditional semiconducting quantum dots, CNDs display remarkable biocompatibility, robust interfacial structure, adequate dispersion in various solvents, and tunable band gap [6]. Indeed, CNDs with tunable fluorescence emissions are considered green nanomaterials that might replace fluorescent semiconductor nanocrystals, which include harmful heavy metals such as Cd [7,8].

Several prior efforts have been devoted to synthesizing CNDs for small- and large-scale production, including the approaches containing the pyrolysis or carbonization of precursors [9,10]. These synthesis routes usually involve hydrothermal/solvothermal treatment [11,12,13,14], thermal pyrolysis [15,16], and laser ablation [17]. In 2015, Hengwei Lin’s group first prepared carbon dots (CDs) from three different types of isomers (*o*-phenylenediamine, *o*-PD; *m*-phenylenediamine, *m*-PD; and *p*-phenylenediamine, *p*-PD), which display intense and steady green, red, and blue colors of luminescence under ultraviolet-light stimulation [18]. This achievement directs us to the facile preparation of CNDs from the three different isomers. Recently, our team devised an effective hydrothermal approach to carbonize three isomers to produce N-functionalized CNDs [12]. Under the illumination of blue light, the photoluminescence (PL) performance of N-doped CND suspension made from o-PD precursor demonstrates outstanding fluorescence with a high quantum yield (QY: 86%) in water. In our earlier research, we primarily concentrated on using hydrothermal synthesis techniques in order to create polyethylene glycol (PEG)-passivated CND clusters [14,19]. The PEG-coated CND clusters exhibit extremely emissive behavior (QY: > 90% in water and in n-hexane) with the assistance of an aggregate-induced emission (AIE) structure. Due to their ultra-high PL performance, both CND samples can be used as photo-indicators to detect various ions in aqueous solutions.

It has previously been shown that CNDs may be utilized as a fluorescent probe for detecting metal ions with an ultra-low detection limit (ppm level) (e.g., Hg^2+^, Pb^2+^, Fe^3+^, and so on) [17,19,20,21,22,23,24,25,26]. CND fluorescence quenching may be used as an in-situ approach to detect metallic ions in aqueous solutions. This approach is easy to use and lacks the complications of certain traditional techniques such as enzyme-linked immunosorbent assays and chromatography/mass spectrometry [27]. However, few researchers have focused on the use of CND and PEG-coated CND clusters for identifying distinct classes of cations/anions. In particular, there is a critical need for real-time analysis of the chemical composition (or toxicity) of various ionic components within wastewater/drinking water. Accordingly, the present work employed two kinds of CNDs (i.e., CND and CND cluster) as photo probers to detect different ions and a variety of metallic cations/anions in the liquid phase. The CND and CND cluster-based detection technique developed in this work has ultra-high sensitivity and exceptional reliability and can be adopted in the real-time assessment of drinking water quality with ppm-level sensitivity towards certain ions.

## 2. Experimental Section

### 2.1. Hydrothermal Synthesis of CND Samples

The hydrothermal synthesis of N-doped CNDs and PEG-coated CND clusters has been reported in our previous publications [12,19]. First, *o*-PD (1.5 g) and PEG (1.5 g, molecular weight: 100,000) were suspended in 50 mL DI water and stirred with a magnetic bar for 60 min at 150 rpm. Then, the suspension was transferred to a silicon dioxide (quartz)-lined autoclave (volume: 75 mL). The hydrothermal synthesis technique was carried out at 300 °C for 2 h. After cooling to room temperature, the solution containing PEG-coated CND samples was freeze-dried for 72 h at −30 °C. To eliminate any insoluble residuals, the PEG-coated CND samples were thoroughly filtered using a microporous separator (average pore size = 0.02 μm). A 72-h freeze-drying step, followed by the nano-filtration stage, was performed to complete the removal of any residual moisture and to obtain CND nanoparticles with no substantial aggregation. Before conducting any experiments, the as-purified CNDs were suspended in ultra-pure water. The schematic diagram illustrating the preparation of CND powders is available in Appendix A. Currently, the synthesis procedure for CNDs was primarily based on our previous experience, as documented in our publications [12,19]. Furthermore, the estimated yield of CND products is approximately 30 wt.%. In this study, “CND” refers only to the purified PEG-free CND samples. In addition, PEG powders with a molecular weight of 100,000 were labeled as “CND-100k” during preparation. Chemical analysis of CND samples was performed using X-ray photoelectron spectroscopy (XPS, Fison VG ESCA210) with a Mg-K radiation emitter. Here, a multiple Gaussian function was used to deconvolve the C 1 s, N 1 s, and O 1 s spectra. The Nicolet 380 spectrometer was used to collect FT-IR spectra. The morphology of the N-doped CND sample was analyzed using high-resolution transmission electron microscopy (HR-TEM, FEI Talos F200 s). During analysis, an accelerating voltage of 200 kV was applied.

### 2.2. Photoluminescence (PL) Emission of N-Doped CND Samples

High-performance ultrasonic bathing was used to prepare the CND suspensions in various solvents for 0.5 h at room temperature. The photograph of CND fluorescence was taken under a UV irradiation source, where the incident intensity of illumination from the UV light (wavelength: 360 nm) was set at 40 Lux. The CND suspensions (pH = ca. 7) stored at ambient temperature under atmospheric pressure for 2 months, generally showed good stability. A fluorescence spectrometer (Hitachi F-7000 FLS920P) was employed to record the PL emission spectra of CND suspensions at 360 nm excitation. CND suspensions’ quantum yield (QY) was determined in relation to Coumarin (C_9_H_6_O_2_, molecular weight: 146; QY: 73% at 373 nm excitation). Following that, the QY value of the CND suspension was calculated using the following equation [28]:QY = QY_r_ × [(PL area/OD)_s_/(PL area/OD)_r_] × *Փ*_s_^2^/*Փ*_r_^2^(1)

In the above equation, the subscriptions “s” and “r” are used for the CND sample and the reference (Coumarin), respectively. The symbol *Փ* is employed to denote the solvent reflective index. The term “PL area” is used to refer to the spectral area of photoluminescence emission, while “OD” represents the absorbance value.

### 2.3. Ionic Detection by the N-Doped CND Samples

To investigate the ionic detection of N-doped CND samples, stepwise ionic solutions were added to CND suspensions and stirred. The metal ions chosen for this work included Al^3+^, Ba^2+^, Ca^2+^, Co^2+^, Fe^3+^, Fe^2+^, Hg^2+^, K^+^, Mg^2+^, Pb^2+^, and Ni^2+^, which were dissolved from Al(NO_3_)_3_, BaCl_2_, CaCl_2_, CoCl_2_, FeCl_3_, FeCl_2_, HgCl_2_, KCl, MgCl_2_, Pb(NO_3_)_2_, and NiCl_2_, respectively. The anions used in this study involved NO_2_^−^, Cl^−^, Br^−^, I^−^, NO_3_^−^, and HSO_4_^−^, which were obtained from the dissolution of NaNO_2_, KCl, KBr, KI, NaNO_3_, and NaHSO_4_, respectively. Herein, the definition of ppm in aqueous solution could be estimated as milligrams per liter. The above CND solutions were stirred homogeneously and incubated at ambient temperature for 1 min. The detection of metal ions and anions within the CND suspensions was carried out at ambient temperature. The fluorescence spectrometer was utilized to record the PL emission spectra at a wavelength of 360 nm. The evolution of PL intensity (i.e., *F*/*F*_0_) with respect to the ionic concentration was also evaluated, where *F*_0_ and *F* were the PL intensities of CND suspensions in the absence and presence of the ions (at various concentrations), respectively. The optical change was identified using a fluorescence spectrometer and digital images were captured with a digital camera.

## 3. Results and Discussion

The XPS measurement was used to investigate the distribution of functional groups on the N-doped CND samples as well as the surface chemistry. XPS analysis showed that both samples (CND and CND-100k) possessed three elements: C 1 s (*ca*. 282–292 eV), N 1 s (*ca*. 396–405 eV), and O 1 s (*ca*. 530–535 eV). The CND sample’s O/C and N/C atomic ratios were 53.6 and 20.2 at. %, respectively, showing that the pristine CND sample contained substantial amounts of oxidation and amidation. The oxidation and amidation extents of the PEG-coated CND cluster (i.e., CND-100k) were reduced to 39.1 and 3.3 at. %, respectively, after PEG bonding. The reduction happened due to the polymer chain (i.e., H−(O−CH_2_−CH_2_)_n_−OH), which has numerous alkylene groups that tend to cover the O-rich and N-doped CND surface completely. As shown in Figure 1, the distribution of oxygen functionalities in CND and CND-100k samples was examined by decomposing the C 1 s and N 1 s peaks with a multiple Gaussian function. Both samples’ C 1 s spectra were deconvoluted into four peaks [C=C/C–C (*ca*. 284.5 eV), C–N (*ca*. 285.8 eV), C–O (*ca*. 286.2 eV), and C=O (*ca*. 287.2 eV)], as shown in Figure 1a,b [28,29]. Figure 1c,d depicts the N 1 s peaks decomposed into three major peaks located at 399.6 eV (pyrrolic or pyridinic N), 400.4 eV (quaternary N), and 401.5 eV (N-oxides) [30,31], where the first and second components can be assigned to the presence of aromatic C=N–C and tertiary N–(C)_3_ bonds, respectively [32,33]. The presence of these two N doping types, so-called “lattice N”, demonstrates that the N-doping onto the CND lattices has been successfully developed. Indeed, the fractional ratio of each functional group is very similar for both CND samples. However, the surface oxidation and amidation levels on the CND-100k samples show a significant decrease after the formation of PEG-coated CND clusters. 

Further, the distribution of surface functional groups was evaluated through FT-IR spectra (Figure 2a) for both samples (CND and CND-100k). For both spectra, there was a broadband peak at around 2800 to 3600 cm^−1^, which is assigned to the typical vibration of N–H, C–H, and O–H bonds [34]. Strong bands at approximately 1620 and 1500 cm^−1^ are ascribed to C=N/C=O and C=C stretching vibrations, respectively, in the carbon structure [35]. Next, a transmittance peak between 1150 and 1250 cm^−1^ corresponds to the C–O stretch of –COOH, indicating the presence of carboxyl and hydroxyl functional groups on CND-100k samples [36]. The XPS analysis and the appearance of the C–O groups originating from the PEG surface layer are in good agreement. In addition, the analysis of the FT-IR spectra verifies that both samples contain nitrogen and oxygen functional groups affixed to the edge or basal plane of CNDs.

Figure 2b displays the HR-TEM micrographs of the CND-100k sample. The formation of PEG-coated clusters was enabled by the well-dispersed nature of the CNDs in the solution, as shown in Figure 2b. Despite the CNDs’ high tendency to aggregate, their dispersion within the solution enables their formation. The formed CND nanoparticles exhibit a particle size distribution that is relatively narrow, with an average diameter in the range of 3–5 nm. Taking into account the polymerization formation of the CND based on phenazine morphology, it is believed that the geometry of the ladder phenazine structure influences the formation of CNDs during the hydrothermal process. Thus, the isomers of phenylenediamine combine to produce mono-bonds. The mono-bonds and amino groups subsequently endure an oxidative process to generate the phenazine structure. Both amino groups may be implicated in the formation of a phenazine-like ladder structure in an oxygen-rich environment [37,38]. Two interwoven polyaniline chains may represent the ladder. This process has also been verified by mathematical models based on the phenylenediamine isomer [39,40]. Consequently, the N-doped CNDs form a 2D structure with modest lateral dimensions and well-defined edge groups [41]. As a result, the ordering of N functionalities might be precisely controlled by modifying the carbon- and nitrogen-containing isomer precursors.

A well-resolved lattice spacing distance of 0.21 nm corresponding to the (100) facet of graphite [42] can be clearly observed in the inset of Figure 2b. Such refined lattice spacing indicates the production of poly-crystalline or amorphous CNDs. The existence of circular rings inside the selected area diffraction (SAD) pattern (see the inset of Figure 3) demonstrates that the single CND powder is of the polycrystalline domain [19]. This finding demonstrates that the in situ hydrothermal procedure is a simple technique to create spherical CNDs as well as PEG-coated clusters during the one-pot synthesis process. We observed a similar trend in the PEG-coated CND clusters in our prior studies [14,19].

The effect of the fluorescence response of the as-prepared CND suspensions exposed to various anions was also evaluated, as shown in Figure 3. Herein, the initial concentration of each ionic suspension was set at 100 ppm. Initially, the QY of CND suspension reaches as high as 86% under UV irradiation. The PL emission spectra of CND and CND-100k suspensions in water induced at 360 nm, which displayed a quasi-symmetric peak with a small tailing between 650 and 700 nm. Under UV illumination, the PL response exhibits a single typical band at ca. 570 nm. The excitation-dependent emission behavior of CND and CND-100k samples has been systematically investigated in previous studies [12,19]. The photostability of CND and CND-100k samples was evaluated by incubating both samples for 2 weeks. We found that the PL intensity variations for both samples were less than 4.9%, indicating the remarkable photostability of CND samples. After exposure to different anions, the QY values vary with the anion type, confirming the possibility of ionic recognition in the liquid phase. The maximal PL quenching of CND suspension can be realized in the presence of the HSO_4_^−^ anion (i.e., the QY value decreases from 86 to 10%). We observed that the CND suspension tends to be transparent after adding the HSO_4_^−^ anion. The PL quenching spectra of CND and CND-100k suspensions vary when different anions are used (see Figure 4a,b). Here, the concentration of the CND suspension was set at 100 ppm, and the concentration of each anion was 0.01 M. As clearly illustrated in Figure 4, both suspensions were quenched with different anions; however, the extent of PL quenching varies, where the presence of HSO_4_^−^ anion has a critical effect on the quenching behavior of both of the suspensions. This finding implies that both the CND suspensions prepared in this work are capable of ionic recognition toward specific anions in the liquid phase.

The fluorescence quenching of the as-prepared suspensions by the HSO_4_^−^ anion was examined through the quantitative detection of the HSO_4_^−^ loading. The PL quenching measurement was carried out by stepwise HSO_4_^−^ addition, where the maximal weight ratio of anion to CND, *R*_M_, was set at 1. Figure 4c,d reveals that the extent of PL quenching strongly depends on the HSO_4_^−^ concentration, where the PL intensities of both CND suspensions decrease with a gradual increase in HSO_4_^−^ concentration (i.e., *R*_M_). The PL quenching of both sensing materials (i.e., CND and CND-100k) by HSO_4_^−^ ions can be quantified by the Stern–Volmer analysis, as described in the following [43,44,45]:*F*_0_/*F* = 1 + *K*_SV_ [*C*](2)

In Equation (2), *F*_0_ and *F* are the PL intensities of CND suspensions in the absence and presence of different ionic species, respectively; *K*_SV_ is the Stern–Volmer quenching constant; and [*C*] is the concentration of the quencher. Importantly, the Stern–Volmer of PL quenching on CNDs by HSO_4_^−^ anions varies linearly with the ionic concentration in the low quencher concentration regime. Thus, the *K*_SV_ value can be determined from the slope of Stern–Volmer linear plots. The limit of detection (LOD) is also a crucial index in evaluating the lowest corresponding quantity of the specific ions in aqueous solution. Employing statistical analysis, the LOD can be expressed using Equation (3) [46]:LOD = 3 σ/*K*_SV_(3)
where σ is the standard deviation of the calibration curves and “3” indicates the confidence level of 95%. The plot of *F*_0_/*F* versus [*C*] was linear (within the entire range of ionic concentrations considered in this work) with a very high coefficient of determination (*R*^2^ > 0.990). The *K*_SV_ value of the CND-100k sample was found to be ca. 3 times higher than that of the CND one (i.e., *K*_SV_ value: 0.062 ppm^−1^ for CND-100k and 0.021 ppm^−1^ for CND). The LODs were also quantified to be 0.57 and 0.19 ppm for the CND and CND-100k samples, indicating that both CND samples are indeed sensitive probes for the detection of HSO_4_^−^.

It is also necessary to analyze the selectivity of the as-prepared CND samples toward various anions present in an aqueous solution [20]. Various anions were added to an aqueous solution, and the difference in PL intensity before and after the addition of HSO_4_^−^ was measured (Figure 5). The selectivity and PL intensities of CND and CND-100k solutions polluted with various anions are shown in Figure 5a,b, respectively. The inhibition of the PL response of CND samples exposed to various anions demonstrates superior selectivity towards HSO_4_^−^ ions, as CNDs undergo significant spectral changes in the presence of I^−^, NO_2_^−^, Br^−^, Cl^−^, and NO_3_^−^ anions. This reveals that the N-doped CND sample has ultra-high sensitivity and excellent selectivity toward HSO_4_^−^ ions among all the anions selected in this study.

The N-doped CND suspensions are also capable of detecting metal ions in the liquid phase. Therefore, the as-prepared CND samples were exposed to various metal ions and their corresponding PL emission spectra were recorded, as displayed in Figure 6. We observe that three metal ions, Fe^3+^, Hg^2+^, and Fe^2+^, can be considered specific quenchers due to their significant PL quenching extent compared to other ionic species. The other N-doped or N-/S-co-doped CNDs, prepared by the solid-phase microwave-assisted technique [20], also exhibit specific detection towards Hg^2+^ ions in the liquid phase. This observation reveals that both CND samples can serve as high-performance sensing probes in ionic recognition. Concerning the specific detection of Hg^2+^ by the N-doped CNDs, the PL quenching mechanism includes two pathways: (i) the formation of an adsorbed complex (i.e., (C*_x_*O)_2_Hg^2+^) in the presence of oxygen functionalities and (ii) a strong interaction between Hg^2+^ ions and amino- and amido-carbonyl functionalization at the edges of graphene sheets (Appendix A). Accordingly, the formation of surface intermediates inhibits photon rejection from the highest occupied molecular orbital gap to the lowest unoccupied molecular orbital gap, resulting in PL quenching behavior.

To further investigate the sensitivity towards Fe^3+^, Fe^2+^, and Hg^2+^ ions, the PL spectra of CND and CND-100k suspensions were analyzed as a function of ionic concentration (see Figure 7, Figure 8 and Figure 9). It is observed that the intensity of the PL reduces as the ionic concentration increases. Further, quenching behavior can be well described by the Stern–Volmer analysis. The *K*_SV_ and LOD parameters along with the *r*^2^ values of both CND samples towards three metal ions were collected and quantified (see Appendix A). As tabulated in Appendix A, with the fitting of the Stern–Volmer equation (i.e., *r*^2^: 0.992–0.999), the *K*_SV_ and LOD parameters were assessed. According to Appendix A, where the *K*_SV_ values are compared, the CND suspension is favorable for Fe^3+^ and Fe^2+^ detection, while Hg^2+^ sensing can be precisely performed by the CND-100k probe. This difference can be attributed to the fact that the PEG-coated CNDs enable the formation of (C*_x_*O)_2_Hg^2+^ due to a higher surface fraction of oxygen functionalities from the PEG skin layer. Appendix A provides a summary of the comparison between the detection limits of Fe^3+^, Fe^2+^, and Hg^2+^ ions using different detectors that were prepared through various synthesis methods [47,48,49,50,51]. As shown in this table, the CND suspension is a highly sensitive fluorescent probe for detecting Fe^3+^ in aqueous solutions at the ppb level. It should also be noted that considering the sensitivity of commonly used precious metal nano-structured probes (e.g., Ag and Au), the sensitivity of the CND-based probes for the Fe^2+^ and Hg^2+^ detection must be further improved. To increase the sensitivity towards Fe^2+^ and Hg^2+^ ions, a comprehensive study is required to determine the optimal detection environment, including chemical composition, solid content, and pH value in batch systems.

Figure 10 illustrates the investigation into the selectivity of CND and CND-100k samples towards Fe^2+^ and Hg^2+^ ions in the presence of different metal ions. We observe that the PL intensities of both CND samples significantly vary in the presence of Fe^2+^ and Hg^2+^ ions. This trend also indicates enhanced selectivity towards specific ionic species on both CND samples within the aqueous solutions.

Appendix Aa is a schematic depiction of several electronic transitions on the CND sample. Individual transitions are shown in the energy band diagram by arrows, indicating a structure characterized by numerous chromophoric band gaps [52,53]. Doping N atoms results in the formation of extra “defect sites”, which introduces additional energy levels and, as a result, creates novel electron transition pathways in the inter-band structure. When exposed to UV light, the absorption of UV photons by the localized electron in double bonds (mostly C=C) produces an electron–hole pair upon electron transition (i.e., excitation). The excited electron is often prone to an intra-band transition from a higher to a lower conduction band, emitting visible light through radiative recombination [28]. This inter-band structure explains why both materials exhibit an asymmetric PL emission pattern when exposed to 360 nm UV light. When PL emission is suppressed by metal ion adsorption (e.g., Fe^3+,^ Fe^2+^, and Hg^2+^), the pathways generated by UV photons are partially or totally blocked by surface intermediates, preventing photon rejection from the inter-band, as shown in Appendix Ab. Within the whole concentration range, the observed PL intensity decreases as the concentrations of Hg^2+^, Fe^2+^, and Fe^3+^ ions increase. This PL quenching of N-doped CNDs by metal ions can be explained by the formation of a CND-(M)*_x_* complex on the carbon surface. Since CNDs contain a large number of oxygen functionalities, the hydrophilic surface makes metal ions in the aqueous solution more accessible to the carbon surface, resulting in greater surface coverage and more effective adsorption. The alteration of surface charge characteristics dramatically modifies the surface–ion interaction at the edge or basal plane of functionalized CNDs. As a result, the development of a large number of surface oxygen groups, such as carboxyl, phenol, and lactone, promotes the interaction of the carbon surface with the mercury ions during the adsorption process. Therefore, with a high concentration of metal ions, higher PL quenching behavior becomes more evident.

## 4. Conclusions

Herein, we established an effective hydrothermal technique for producing CND and PEG-coated CND nanomaterials as extremely sensitive PL probes for detecting particular anions and metallic ions in aqueous solutions. In this study, both CND and CND-100k suspensions were shown to be excellent sensing probes for ionic recognition in the liquid phase. Through the PL quenching, CND and PEG-coated CND suspensions demonstrated superior sensitivity and selectivity toward HSO_4_^−^ anions with a ppm-level detection limit in the liquid phase. The quenching mechanism of N-doped CNDs towards HSO_4_^−^ ions can be described in two steps including (i) bidentate H-bonding and (ii) monodentate H-bonding with sulfate anionic moieties. Considering the various metallic ions usually found in aqueous solutions, the Stern–Volmer analysis revealed that the CND suspension is appropriate for the detection of Fe^3+^ and Fe^2+^ ions, whereas Hg^2+^ sensing can be mainly performed by the PEG-coated CND clusters. Indeed, the as-prepared CND suspensions are powerful PL sensing probes for detecting a variety of anions and metal ions commonly found in aqueous media. This work’s ionic sensing framework paves the way for the development of extremely sensitive and selective probes for detecting diverse target ions in aqueous solutions.

## Figures and Tables

**Figure 1 polymers-15-02655-f001:**
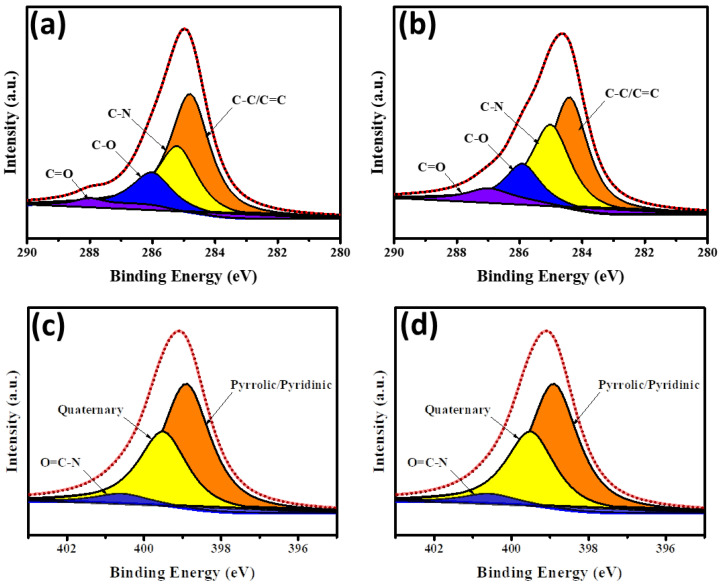
High-resolution XPS C 1 s spectra of (**a**) CND and (**b**) CND-100k samples and XPS N 1 s spectra of (**c**) CND and (**d**) CND-100k samples.

**Figure 2 polymers-15-02655-f002:**
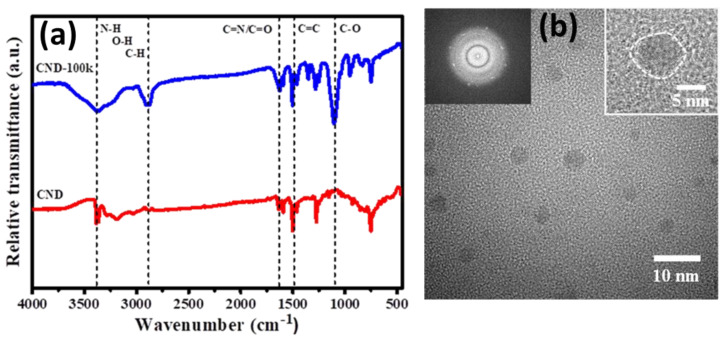
(**a**) FT-IR spectra of CND and CND-100k samples; (**b**) HR-TEM micrograph of CND-100k sample, where black dots represent CNDs covering over PEG polymer. The inset shows SAD pattern (**left**) and lattice fringes in individual CND powder (**right**).

**Figure 3 polymers-15-02655-f003:**
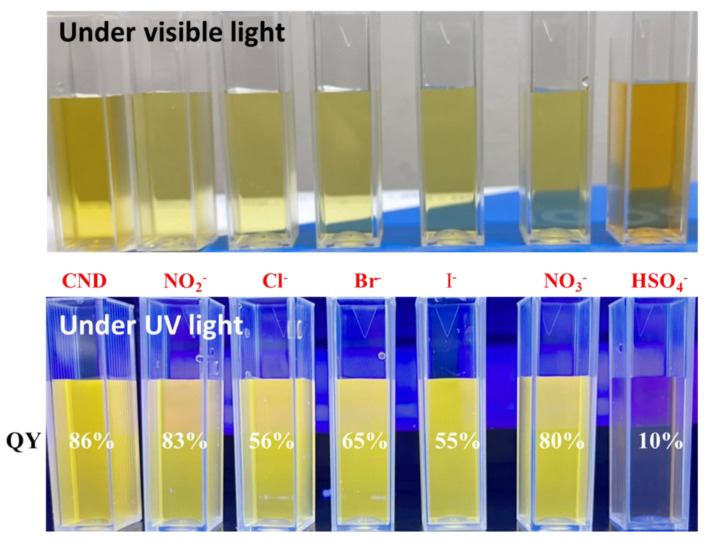
Photographs of CND suspensions before (**top**) and after adding different anions under UV irradiation (**bottom**). The initial concentration of each ionic suspension was set at 100 ppm.

**Figure 4 polymers-15-02655-f004:**
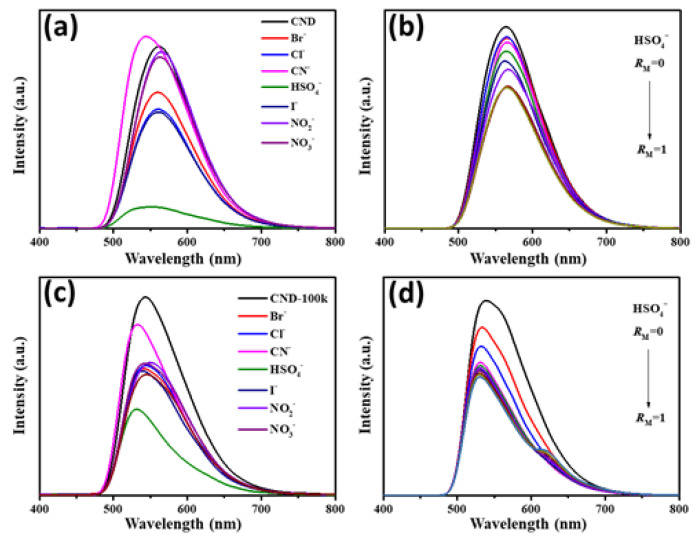
PL emission spectra of CND suspensions (**a**) after adding different anions and (**b**) varied with different HSO_4_^−^ concentrations. PL emission spectra of CND-100k suspensions (**c**) after adding different anions and (**d**) varied with different HSO_4_^−^ concentrations.

**Figure 5 polymers-15-02655-f005:**
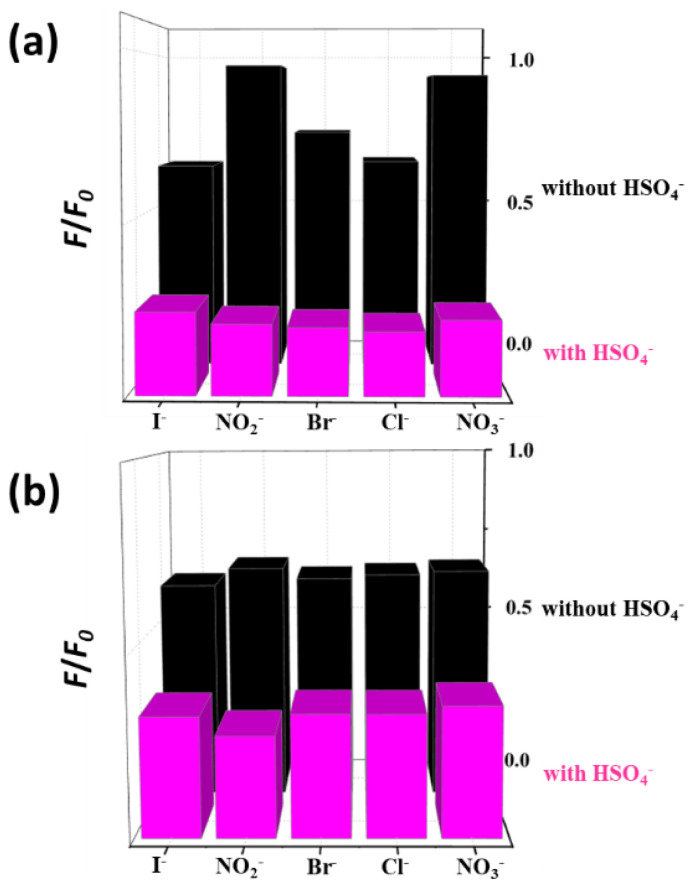
PL intensity ratio as a function of anion type before and after adding HSO_4_^−^ ions: (**a**) CND and (**b**) CND-100k suspensions.

**Figure 6 polymers-15-02655-f006:**
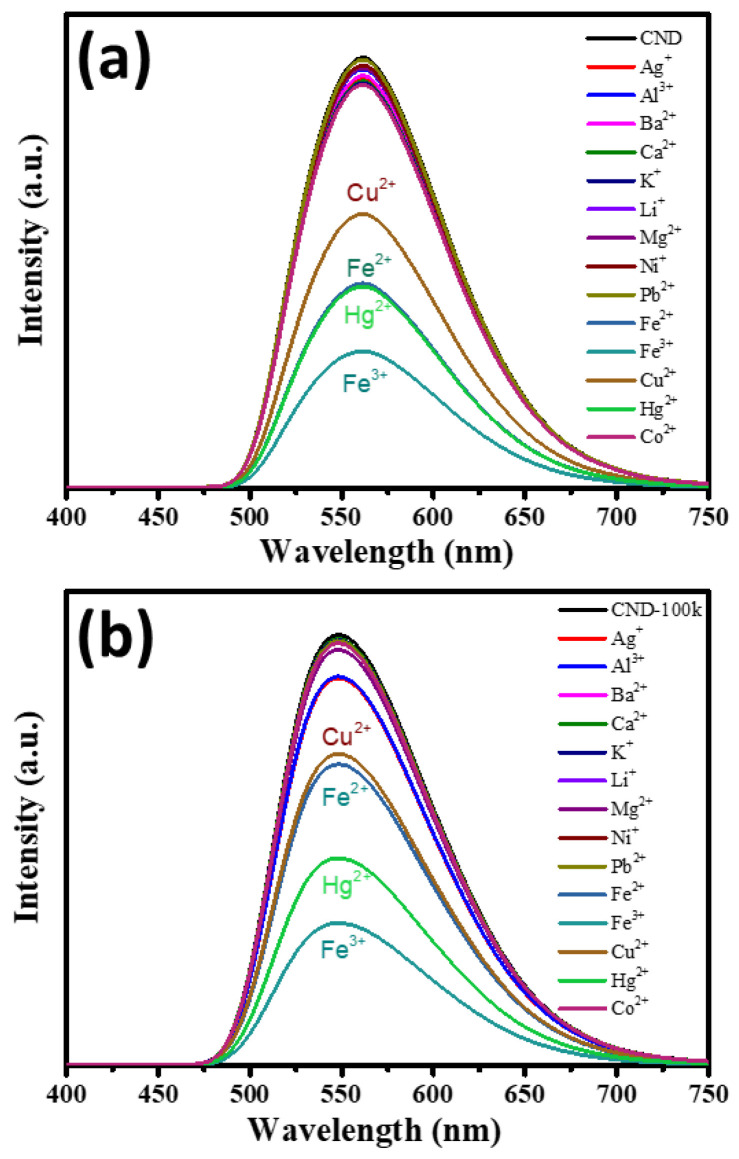
PL emission spectra from (**a**) CND and (**b**) CND-100k suspensions after adding different metal ions.

**Figure 7 polymers-15-02655-f007:**
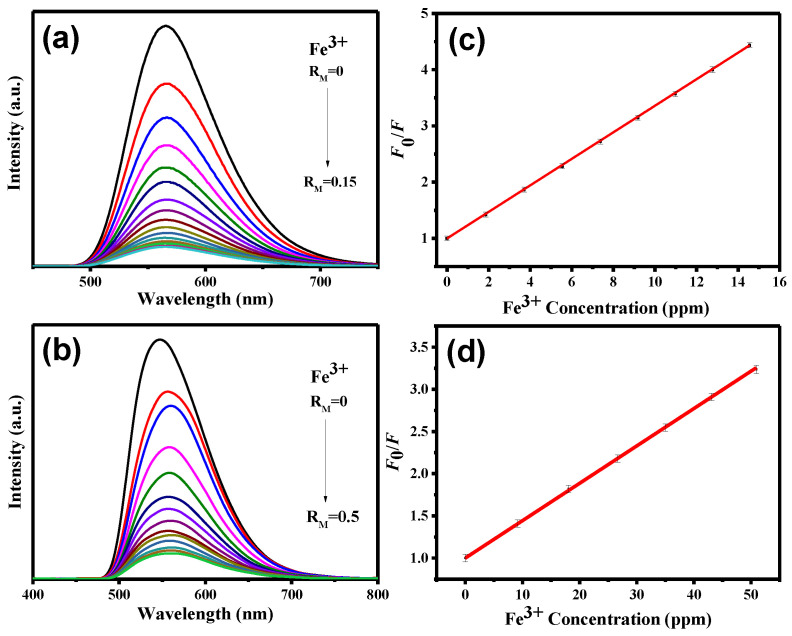
PL emission spectra from (**a**) CND and (**b**) CND-100k suspensions after adding different amounts of Fe^3+^ ionic solution. Stern–Volmer plot of PL intensity ratio versus Fe^3+^ concentration at 360 nm: (**c**) CND and (**d**) CND-100k suspensions.

**Figure 8 polymers-15-02655-f008:**
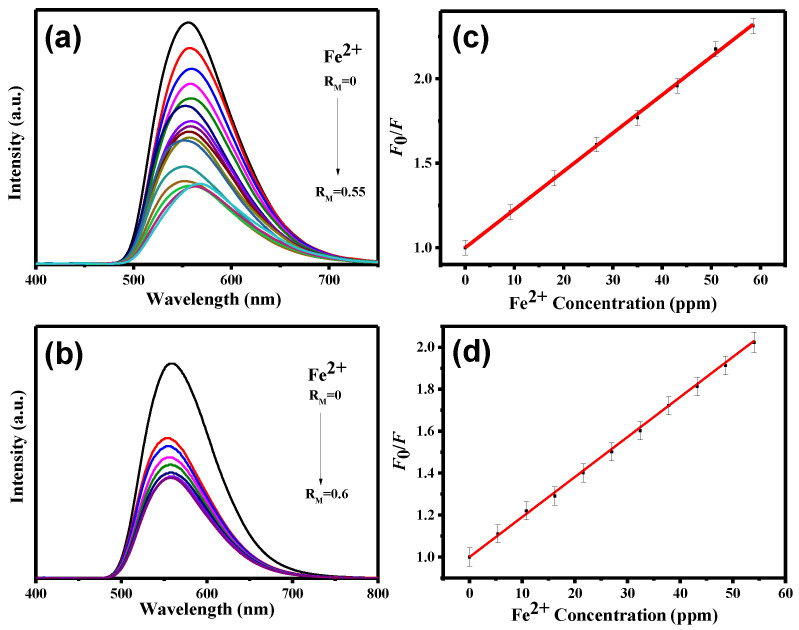
PL emission spectra from (**a**) CND and (**b**) CND-100k suspensions after adding different amounts of Fe^2+^ ionic solution. Stern–Volmer plot of PL intensity ratio versus Fe^2+^ concentration at 360 nm: (**c**) CND and (**d**) CND-100k suspensions.

**Figure 9 polymers-15-02655-f009:**
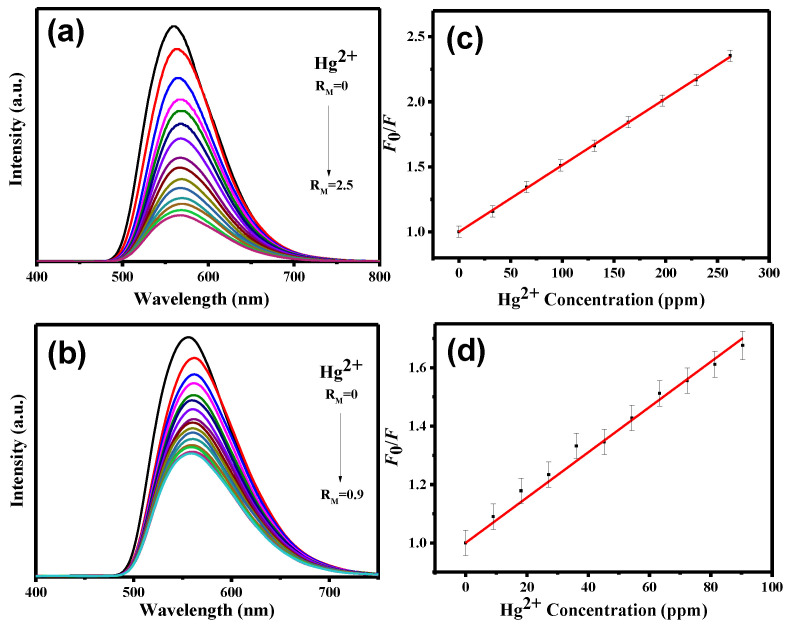
PL emission spectra from (**a**) CND and (**b**) CND-100k suspensions after adding different amounts of Hg^2+^ ionic solution. Stern–Volmer plot of PL intensity ratio versus Hg^2+^ concentration at 360 nm: (**c**) CND and (**d**) CND-100k suspensions.

**Figure 10 polymers-15-02655-f010:**
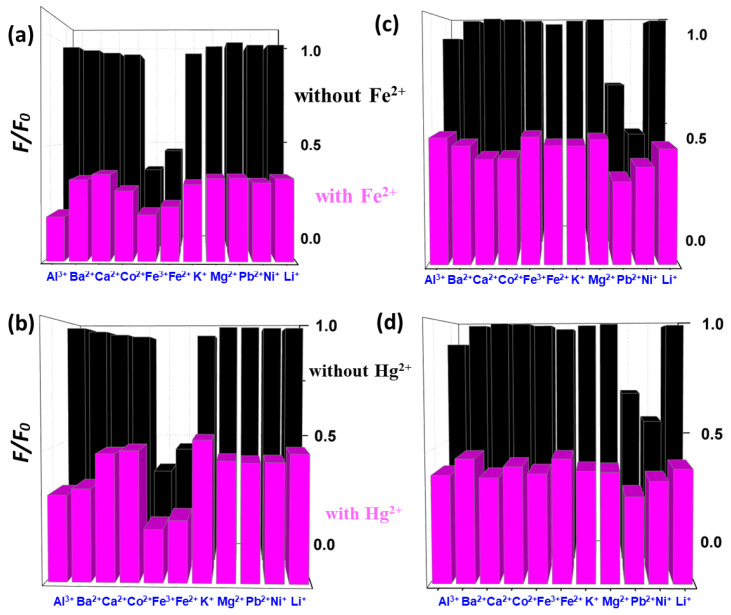
PL intensity ratio of CND suspension varied with different metal ions in the presence of (**a**) Fe^2+^ and (**b**) Hg^2+^ ions. PL intensity ratio of CND-100k suspension varied with different metal ions in the presence of (**c**) Fe^2+^ and (**d**) Hg^2+^ ions.

## Data Availability

Data are contained within the article or Appendix A.

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
