# Peer review of "Hydrothermal Synthesis of Functionalized Carbon Nanodots and Their Clusters as Ionic Probe for High Sensitivity and Selectivity for Sulfate Anions with Excellent Detection Level"

_polymers, 2023, doi:10.3390/polym15122655_

Round 1

Reviewer 1 Report (Previous Reviewer 2)

Comments: This article reports the hydrothermal synthesis of functionalized carbon nanodots and their clusters as ionic probe for high sensitivity and selectivity for sulfate anions with excellent detection level. The structure of the synthesized materials has been characterized well and these analyses are reasonable. However, authors should address the following comments for its acceptance.

1.    Abstract must be improved with some numerical results drawn from the studies.

2.    Numerous studies on similar materials have been reported in the literature, how your work is different from published work? What is the novelty of the present work? Why was N selected as heteroatoms? How does doping enhance performance? All these points should be explained in the introduction part.

3.    What about the excitation-dependent emission behavior of carbon dots, authors can discuss these properties for the understanding of readers.

4.    What about the photostability of the prepared CNDs?

5.    In order to show the superiority of the current materials, comparisons over the other related materials reported in the past literatures are necessary. Sensing performances of the current materials have to be compared with those of the other materials and reasons for performance improvements have to be discussed.

6.    The authors must double-check the whole manuscript to eliminate syntax and format errors.

Minor grammatical checks are required.

Author Response

  1. Abstract must be improved with some numerical results drawn from the studies.

Ans: Page 1. The reviewer’s suggestion was adopted. The Abstract of the revised manuscript has been moderately modified, where the breakthrough of this study has been pointed out. We have reflected them in the revised manuscript.

  1. Numerous studies on similar materials have been reported in the literature, how your work is different from published work? What is the novelty of the present work? Why was N selected as heteroatoms? How does doping enhance performance? All these points should be explained in the introduction part.

Ans: We adopted the reviewer’s suggestion to emphasize the novelty of the present work in the revision. We also pointed out why we selected N as heteroatoms and how the N dopant improves the PL performance in the Introduction part.

  1. What about the excitation-dependent emission behavior of carbon dots, authors can discuss these properties for the understanding of readers.

Ans: The reviewer’s concern is appropriate. For good readability, the excitation-dependent emission behavior of carbon dots has been reflected in the revised manuscript.

  1. What about the photostability of the prepared CNDs?

Ans: Page 6. The photostability of the prepared CND suspensions can be stored for two weeks and their PL emissions just displayed a slight decay < 4.9 % under UV illumination (i.e., 360 nm). One brief description has been added into the revised manuscript.

  1. In order to show the superiority of the current materials, comparisons over the other related materials reported in the past literatures are necessary. Sensing performances of the current materials have to be compared with those of the other materials and reasons for performance improvements have to be discussed.

Ans: Page 9. The comparisons of PL sensing performance between this study and the past literatures have been collected and listed in the Electronic Supporting Information (see Table S2). We have briefly discussed about the comparison in the revised manuscript.

  1. The authors must double-check the whole manuscript to eliminate syntax and format errors.

Ans: The reviewer’s suggestion was adopted. One native speaker has carefully checked the whole manuscript to eliminate syntax and format errors.

Reviewer 2 Report (Previous Reviewer 1)

This manuscript describes the hydrothermal synthesis of CNDs and their performance to detect specific ions. The manuscript is well-organized and could be accepted after addressing several minor issues.

1. The topic of this manuscript is the application of CNDs as ionic probes, whereas only 3 sentences cover the current status of this application. Some recent progress are suggested to be included in the introduction section, such as Anal. Chem. 2023, 95, 19, 7584; and Nano Energy, 101 (2022) 107549.

2. Raw data of the XPS spectra are suggested to be added in Figure 1 for better evaluation of the fitting results.

3. Several figures such as Figures 2 and 3 are suggested to be integrated together to make the manuscript more concise.

4. Figure 11 should be replotted because the details of this figure can hardly be recognized.

Good.

Author Response

  1. The topic of this manuscript is the application of CNDs as ionic probes, whereas only 3 sentences cover the current status of this application. Some recent progress is suggested to be included in the introduction section, such as Anal. Chem. 2023, 95, 19, 7584; and Nano Energy, 101 (2022) 107549.

Ans: The crucial papers mentioned by the reviewer have been cited at their appropriate locations of the revised manuscript. For better readability, one brief description has been added into the revision.

New references:

Anal. Chem. 2023, 95, 19, 7584. (i.e., reference 26)

Nano Energy, 101 (2022) 107549. (i.e., reference 25)

  1. Raw data of the XPS spectra are suggested to be added in Figure 1 for better evaluation of the fitting results.

Ans: We adopted the reviewer’s suggestion to add the raw data to XPS spectra, as illustrated in Figure 1. We have reflected it in the revised manuscript.

  1. Several figures such as Figures 2 and 3 are suggested to be integrated together to make the manuscript more concise.

Ans: We agree with the reviewer’s opinion. In the revised manuscript, Figures 2 and 3 have been integrated together, as shown in the revised manuscript.

  1. Figure 11 should be replotted because the details of this figure can hardly be recognized.

Ans: The reviewer is right. To avoid reader’s confusion, original Figure 11 has been replotted with high resolution (Now it is Figure 10). We have reflected it in the revision.

Reviewer 3 Report (New Reviewer)

Overall, the presented study is a nice piece of research in the field of CND. Investigation of CND is among the highest topics in modern nanoscience, and more than 10 papers are published daily in this field. We have recently prepared a MSc paper with my student, and I can say that the manuscript by Yang et al. is well above average. Even though it does not present a new concept or generalization, it is a well planned and accurately performed piece of research. I suggest publishing it in Polymers upon considering several issues and questions listed below. 

1. Please remove 'novel' from the abstract (line 14). The procedure described in the paper may be regarded as a variation of known procedures (for example, these from refs. 14, 18, and, I believe, many others) - but I would not call it 'novel'.

2. Actually hydrocarbons are not the most common precursors of CND (as it is stated in lines 31 and 42). I would not explicitly mentioned them (or, at least, add other examples like carbohydrates, acids, amines, etc).

3. Line 44: please change 2005 to 2015.

4. I do not quite understand the piece in lines 75-77:

"The hydrothermal synthesis of N-doped CNDs has been reported in our previous 75 publication [12]. In short, the procedure of PEG-coated CND clusters can be briefly de-76 scribed as follows [19]."

Where was the synthesis method adopted from - ref. 12 or 19? Is there a need to cite both?

5. Regarding the sample preparation and isolation (lines 77-86): how do you know that the CND were purified of the low-molecular admixtures? Why did not you use dialysis as is common about CND synthesis and purification? Did you try to estimate the yield of CND in the reaction? Is PEG stable under the hydrothermal conditions?

6. Line 122. 'Fluctuation' normally refers to some random variation. I would replace it with 'change' or 'evolution' or whatever else appropriate.

7. Regarding the experiments with sensitivity (lines 112-126) - please explain the conversion between ppm and common concentration units. ppm can be understood as mol ions per mol solvent, or mol ions per mol of total species in solution, or g per g solution, or even mol per g. Now it is not quite convenient to compare the obtained results with these given in other studies.

8. Line 136. I do not quite understand, why 'alkyl groups tend to cover O-rich and N-doped surface'. Moreover, strictly speaking CH2CH2 groups are not alkyl but rather alkylene ones.

9. Lines 158, 160: please give the bond explicitly, i.e. C-O instead of CO which can be understood differently.

10. In Fig. 2 and the related discussion - are you aware of the assignment of the band at 1250 1/cm, which is quite strong?

11. Line 170. I think, something like 'monodisperse' or 'narrow' is more appropriate to describe the distribution than 'homogeneous'. "Homogeneous distribution' sounds like "uniform", which is a different story.

12. Figure 4 and the related discussion in lines 196-205: please report the concentration of salts used in this qualitative experiment.

13. Fig. 4 - what does "85%" to the right of the bottom-lane images mean?

14. Line 221 and Fig. 5 - maximal weight ratio is said to be 0.2 in the text but 1 is given in the figure.

15. Did you check that the effect of the addition of sodium hydrosulfate is indeed due to the interaction with the anion, not the effect of the medium pH? (pH of the NaHSO4 solution seems to be well below 7, in contrast to other salts reported in Fig. 4.

16. Fig. 4 - the CND are labeled as 'GND' in the figure, please make it consistent.

17. Line 237. I suggest to use "correlation coefficient" or "coefficient of determination" instead of "correlation factor"; it seems to me that the latter term is less common.

18. Line 257. Please remove Zn2+ from the list of anions. Please make 2 and 3 in nitrite and nitrate subscripts instead of superscripts.

19. Line 263. I would replace 'metallic' with 'metal'.

20. Regarding the discussion about sensitivity to cations. Did you consider the fact that Fe(III) strongly absorbs UV light, and its effect can be due to the inner filter rather than other mechanisms?

21. Line 317. It seems that it should be 'species', not 'spices'.

Author Response

  1. Please remove “novel” from the abstract (line 14). The procedure described in the paper may be regarded as a variation of known procedures (for example, these from refs. 14, 18, and, I believe, many others) - but I would not call it “novel”.

Ans: Page 1. We agree with the reviewer’s opinion. The inappropriate word “novel” has been eliminated from original manuscript. We have reflected it in the revision.

  1. Actually hydrocarbons are not the most common precursors of CND (as it is stated in lines 31 and 42). I would not explicitly mention them (or, at least, add other examples like carbohydrates, acids, amines, etc).

Ans: Page 1. The reviewer’s concern is appropriate. The sentence has been moderately modified according to the reviewer’s suggestion.

  1. Line 44: please change 2005 to 2015.

Ans: Page 2. The wrong number (2005) has been corrected to 2015. We have reflected it in the revised manuscript.

  1. I do not quite understand the piece in lines 75-77.

Ans: Page 2. In order to prevent reader confusion, we have combined the two sentences together. Specifically, we have mentioned in our previous publications the hydrothermal procedure for N-doped CNDs and the use of PEG-coated CND clusters. We have reflected it in the revised manuscript.

  1. Regarding the sample preparation and isolation (lines 77-86): how do you know that the CND were purified of the low-molecular admixtures? Why did not you use dialysis as is common about CND synthesis and purification? Did you try to estimate the yield of CND in the reaction? Is PEG stable under the hydrothermal conditions?

Ans: Page 2. The reviewer’s concern is valid. The dialysis technique is indeed one of the most effective methods for purifying nanoparticles. However, we lack information regarding the duration of the separation period and the particle size range suitable for this technique. Currently, the synthesis procedure for CNDs is primarily based on our previous experience, as documented in our publications. Furthermore, the estimated yield of CND products is approximately 30 wt.%. A brief description addressing these points has been included in the revised manuscript.

  1. Line 122. “Fluctuation” normally refers to some random variation. I would replace it with “change” or “evolution” or whatever else appropriate.

Ans: Page 3. In order to provide a more suitable description, we have replaced the term “fluctuation” with “evolution” in the revised manuscript, as suggested by the reviewer.

  1. Regarding the experiments with sensitivity (lines 112-126) - please explain the conversion between ppm and common concentration units. ppm can be understood as mol ions per mol solvent, or mol ions per mol of total species in solution, or g per g solution, or even mol per g. Now it is not quite convenient to compare the obtained results with these given in other studies.

Ans: Page 3. We have incorporated the suggestion made by the reviewer. The revised manuscript now includes the definition of ppm in aqueous solution, which can be estimated as milligrams per liter.

  1. Line 136. I do not quite understand, why “alkyl groups tend to cover O-rich and N-doped surface”. Moreover, strictly speaking CH2CH2 groups are not alkyl but rather alkylene ones.

Ans: Page 3. The reviewer is right. The inappropriate description “alkyl groups…” has been modified. We have reflected it in the revised manuscript.

  1. Lines 158, 160: please give the bond explicitly, i.e. C-O instead of CO which can be understood differently.

Ans: Page 4. We have modified C‒O instead of CO in the sentences, as shown in the revised manuscript.

  1. In Fig. 2 and the related discussion - are you aware of the assignment of the band at 1250 1/cm, which is quite strong?

Ans: Page 4. The reviewer’s inspection was correct. The strong band at approximately 1250 cm-1 can be assigned to the stretching vibration of C‒O groups. We have reflected it in the revision.

  1. Line 170. I think, something like “monodisperse” or “narrow” is more appropriate to describe the distribution than “homogeneous”. “Homogeneous distribution” sounds like “uniform”, which is a different story.

Ans: Page 5. We agree with the reviewer’s opinion to replace term “homogeneous” with “narrow” in the revised manuscript.

  1. Figure 4 and the related discussion in lines 196-205: please report the concentration of salts used in this qualitative experiment.

Ans: Page 6 and 7. The initial concentration of each ionic suspension was set at 100 ppm, as shown in Figure 3. We have reflected it in the caption of Figure 3.

  1. Fig. 4 - what does “85%” to the right of the bottom-lane images mean?

Ans: Figure 3. The typo has been removed from original manuscript.

  1. Line 221 and Fig. 5 - maximal weight ratio is said to be 0.2 in the text but 1 is given in the figure.

Ans: Page 5. We have corrected the maximal weight ratio (= 1.0) in the revised manuscript.

  1. Did you check that the effect of the addition of sodium hydrosulfate is indeed due to the interaction with the anion, not the effect of the medium pH? (pH of the NaHSO4 solution seems to be well below 7, in contrast to other salts reported in Fig. 4.

Ans: The reviewer’s concern is appropriate. The impact of pH value on the photoluminescence (PL) quenching of sodium hydrosulfate can be considered as a potential research topic for future investigations.

  1. Fig. 4 - the CND are labeled as “GND” in the figure, please make it consistent.

Ans: Figure 3. We have corrected the unwitting mistake in Figure 3.

  1. Line 237. I suggest to use “correlation coefficient” or “coefficient of determination” instead of “correlation factor”; it seems to me that the latter term is less common.

Ans: Page 7. We have implemented the suggestion provided by the reviewer. The term “correlation factor” has been revised to “coefficient of determination” in accordance with the reviewer's recommendation.

  1. Line 257. Please remove Zn2+ from the list of anions. Please make 2 and 3 in nitrite and nitrate subscripts instead of superscripts.

Ans: Page 2. We have rectified the unintentional errors and incorporated the necessary corrections into the revised manuscript.

  1. Line 263. I would replace “metallic” with “metal”.

Ans: Page 8. The inappropriate word has been replaced by “metal ions” in the revised manuscript.

  1. Regarding the discussion about sensitivity to cations. Did you consider the fact that Fe(III) strongly absorbs UV light, and its effect can be due to the inner filter rather than other mechanisms?

Ans: In response to the reviewer’s valuable feedback, we acknowledge the significance of the reviewer’s inspection. The reviewer has proposed an intriguing approach to enhance photosensitivity for cation sensing in aqueous solutions. Exploring the potential of intense absorbance of Fe(III) ions in improving cation sensitivity could be pursued as a promising avenue for future research. However, it is important to conduct a comprehensive investigation, including studying the UV absorbance in the absence of CNDs. We express our gratitude to the reviewer for the reviewer’s valuable and meaningful suggestions.

  1. Line 317. It seems that it should be “species”, not “spices”.

Ans: Page 11. The unintentional error has been corrected in the revised manuscript.

Round 2

Reviewer 1 Report (Previous Reviewer 2)

The revised version of the manuscript may acceptable to the journal standard.

This manuscript is a resubmission of an earlier submission. The following is a list of the peer review reports and author responses from that submission.

Round 1

Reviewer 1 Report

The authors present the synthesis of N-doped Carbon nanodots and demonstrated their potential application as ion indicators. The manuscript is kind of interesting. However, some important issues need to be fully addressed.

1. What is the origin of the selectivity on various ions and cations? Why some ions exhibit significant quenching effect while others don't?

2. Why the PL intensity decreases linearly with the ion concentration? Is that taken through tuning the absorption? Or quantum efficiency? More detailed study should be carried out.

Author Response

1. Page 8. The reviewer’s concern is appropriate. As for the specific detection of mercury ions by the N-doped CNDs, the PL quenching mechanism includes two pathways: (i) formation of adsorbed complex (i.e., (CxO)2Hg2+) in the presence of oxygen functionalities and (ii) strong interaction between Hg2+ ions and amino- and amid-carbonyl functionalization at the edges of graphene sheets. Accordingly, the formation of surface intermediates effectively hinders the photon rejection from the highest occupied molecular orbital ‒lowest unoccupied molecular orbital gap, leading to PL quenching behavior. However, an in-depth investigation is required to assess the optimal detection environment such as pH value, chemical composition, and the solid content in batch system. We have reflected it in the revised manuscript.

2. Page 12. One brief description regarding the PL quenching by the presence of iron and mercury ions has been added to figure out the quenching mechanism. The measured PL intensity shows a decreasing function of concentration of Hg2+, Fe2+, and Fe3+ ions within the entire concentration ranges. This PL quenching of N-doped CNDs by the metal ions can be interpreted by the formation of a CND‒(M)x complex on the carbon surface. Since the CNDs contain a large number of oxygen functionalities, the hydrophilic surface thus allows better accessibility of metal ions in the aqueous solution to the carbon surface, leading to ample surface coverage and a more efficient adsorption. This significantly alters the surface-ion interaction at the edge or basal plane of functionalized CNDs by the modification of surface charge properties. Thus, the formation of a great number of surface oxygen groups, such as carboxyl, phenol, and lactone, favor the interaction between the carbon surface and the mercury ions during the adsorption process. Therefore, the high concentration of metal ions, the higher PL quenching behavior becomes more evident. We have reflected it in the revised manuscript.

Reviewer 2 Report

Comments: This article reports the hydrothermal synthesis of functionalized carbon nanodots and their clusters as ionic probe for high sensitivity and selectivity for sulfate anions with excellent detection level. The structure of the synthesized materials has been characterized well, and these analyses are reasonable. Authors should address the following comments for its acceptance.

1.    How is the synthesis method different or better than those reported earlier? This points should be explained in the introduction part.

2.    Author needs to supply and compare the survey spectra of CND and CND-100k samples to determine their functionalities and for a clear understanding of readers.

3.    What about the excitation-dependent emission behavior of CND and CND-100k samples, authors can discuss these properties for the understanding of readers.

4.    Auther needs to investigate the photostability of the prepared CND and CND-100k samples.

5.    The authors need to double-check the whole manuscript to eliminate some syntax and format errors.

Author Response

1. Page 2. The reviewer’s suggestion was adopted. We have emphasized the novelty of the synthesis method in the Introduction part. We have reflected it in the revision.

2. Page 3 and 4. The reviewer’s concern is appropriate. We have discussed the difference between CND and CND-100k sample in XPS and FT-IR spectra in the revised manuscript. The XPS measurements confirmed the existence of three main elements: C 1s (ca. 282‒292 eV), N 1s (ca. 396‒405 eV), and O 1s (ca. 530‒535 eV) on both N-doped CND samples, i.e., CND and CND-100k. The O/C and N/C atomic ratios of the CND sample were ~53.6 and 20.2 at.%, respectively, indicating that the pristine CND sample contains high oxidation and amidation levels. Upon PEG bonding, the oxidation and amidation extent of the PEG-coated CND cluster (i.e., CND-100k) decreased to 39.1 at.% and 3.3 at.%, respectively. Such a reduction in the O/C and N/C atomic ratios after the PEG coating is majorly due to the polymer chain (i.e., H−(O−CH2−CH2)n−OH) containing a large number of alkyl groups that tend to totally cover the O-rich and N-doped CND surface. We have reflected it in the revision.

3. Page 6. We adopted the reviewer’s suggestion to provide a brief description in the revised manuscript. As illustrated in Figure 4, initially, the QY of CND suspension reaches as high as 86% under the UV irradiation. The PL emission spectra of CND and CND-100k suspensions in water excited at 360 nm display quasi-symmetric peak associated with a slight tailing under 650‒700 nm. The PL response exhibits a single typical band at ca. 570 nm under UV illumination. The excitation-dependent emission behavior of CND and CND-100k samples has been systematically investigated in previous studies [12,19].

4. Page 6. The reviewer’s comments are valuable. The photostability of the prepared CND and CND-100k samples has been confirmed by storing both samples in aqueous solutions after 2 weeks. The PL intensities of both samples still maintains > 95.1 % of their original values under UV illumination. One brief description has been added into the revised manuscript.

5. The main text. We adopted the reviewer’s suggestion. To improve the quality, a native speaker has revised the main text and corrected the formatting errors. The corrections made by the native speaker have been marked by blue color. We have reflected it in the revised manuscript.

Reviewer 3 Report

Development of a new type carbon nanodots for more applications is an important topic in current field of carbon nano-materials. The authors report the functionalized carbon nanodots and their clusters as ionic probe for sulfate anions. However, both of the sythesis and the applications are lack of innovation. 

1.         The process of synthesis is not novel, the method to carbonize three types of isomers (i.e., o-, m-, and p-phenylenediamine (o-PD, m-PD, and p-PD) in order to form the N-functionalized CNDs (line 44-45) had been reported since 2015 (Jiang K, Sun S, Zhang L, et al. Red, green, and blue luminescence by carbon dots: full‐color emission tuning and multicolor cellular imaging[J]. Angewandte chemie, 2015, 127(18): 5450-5453.).

2.         The PEG-coated GQD samples…? Is it not CNDs?

3.         Today, the QY of CND should not be measured with respect to the Coumarin, the authors should give the absolute QY.

4.         The written of the paper should be given more attention, the Figure1,  2 and 3 have the same notes!!!

In this sense, I feel this work is not novel enough to warrant publication in Polymers.

Author Response

1. Page 1-2. We agree with the reviewer’s opinion. The important reference, mentioned by the reviewer #3, has been cited at its appropriate location of the revised manuscript. One brief description regarding the novel synthesis of CNDs from three types of isomers (i.e., o-, m-, and p-phenylenediamine) has been added into the revision.

New reference:

  1. K. Jiang, S. Sun, L. Zhang, Y. Lu, A. Wu, C. Cai, H. Lin, Red, green, and blue luminescence by carbon dots: full‐color emission tuning and multicolor cellular imaging. Angew. Chem. 127 (2015) 5450-5453.

2. Page 2. The reviewer is right. The unwitting mistake has been corrected. We have reflected it in the revised manuscript.

3. The reviewer’s concern is appropriate. The QY value with respect to the Coumarin is one of the common references in evaluating the PL efficiency of CNDs. Of course, in the development of new light-emitting materials, it is often essential to improve their PL efficiency. This requires accurate techniques for measuring the quantum yield (the ratio of the number of photons emitted through photoluminescence to the number of photons absorbed by the light-emitting material). However, we feel that the PL quenching by the presence of metal ions (Fe2+, Fe3+, and Hg2+) can be accurately measured by the relative PL intensity, as shown in Figures 8, 9, and 10. The absolute QY values can be subjected as another topic in the near future.

4. Page 5 and 6. The reviewer is correct. We have corrected the typos for the captions of Figures 2 and 3. We have reflected them in the revised manuscript.